# The Role of Immunotherapy in the Treatment of Rare Central Nervous System Tumors

Andrew Rodriguez, Carlos Kamiya-Matsuoka and Nazanin K. Majd *

Department of Neuro-Oncology, MD Anderson Cancer Center, Houston, TX 77030, USA;
aerodriguez2@mdanderson.org (A.R.); ckamiya@mdanderson.org (C.K.-M.)
* Correspondence: nkmajd@mdanderson.org

**Abstract:** Establishing novel therapies for rare central nervous system (CNS) tumors is arduous due to challenges in conducting clinical trials in rare tumors. Immunotherapy treatment has been a rapidly developing field and has demonstrated improvements in outcomes for multiple types of solid malignancies. In rare CNS tumors, the role of immunotherapy is being explored. In this article, we review the preclinical and clinical data of various immunotherapy modalities in select rare CNS tumors, including atypical meningioma, aggressive pituitary adenoma, pituitary carcinoma, ependymoma, embryonal tumor, atypical teratoid/rhabdoid tumor, and meningeal solitary fibrous tumor. Among these tumor types, some studies have shown promise; however, ongoing clinical trials will be critical for defining and optimizing the role of immunotherapy for these patients.

**Keywords:** immunotherapy; treatment; rare; central nervous system tumors

## 1. Introduction

Primary malignancies of the CNS—classified by the World Health Organization (WHO) and the Consortium to Inform Molecular and Practical Approaches to CNS Tumor Taxonomy (c-IMPACT NOW)—include almost 150 types of tumors [1]. Many of these tumors are rare, affecting fewer than 1000 patients each year in the United States [2]. In this article we selected the following rare CNS tumors due to their potential immunogenicity: atypical or anaplastic meningiomas, pituitary carcinomas, ependymomas, embryonal tumors, atypical teratoid/rhabdoid tumors, embryonal tumors, and solitary fibrous tumors. These tumors' rarity limits many patients' access to cancer centers with experience in treating rare CNS tumors; likewise, pharmaceutical companies are less eager to invest in research for new treatments for rare tumors. These factors have hampered efforts to conduct clinical studies of rare CNS tumors, which limits patients' treatment options. An increasing body of evidence supports the use of immunotherapy in a number of solid cancers, but the role of immunotherapy in rare CNS tumors remains elusive. Here, we review the use of immunotherapy and preclinical and clinical reports of immunotherapy outcomes in select rare CNS tumors.

*The Landscape of Immunotherapy Modalities*

Immunotherapy has been a rapidly developing field that aims to augment the natural immune defenses to eliminate malignant cells. The main categories of immunotherapy include immune checkpoint inhibitors (ICIs), oncolytic viruses, cancer vaccines, cytokines, and adoptive cell therapies [3]. Below, we briefly describe each modality.

ICIs are the leading immunotherapy to date and are approved by the US Food and Drug Administration (FDA) for the treatment of a number of solid cancers, including melanoma and non-small cell lung cancer. Immune checkpoints are coinhibitory signaling pathway molecules that preserve immune tolerance. However, cancer cells utilize this machinery to evade immunosurveillance. ICIs disable the inhibitory signals to promote T-cell activity for the elimination of cancer cells. Common checkpoint molecules include CTLA-4,

PD-1, and PD-L1. CTLA-4 is expressed on T-cells and downregulates T-cell activation. In a landmark study, the inhibition of CTLA-4 with blocking antibodies was shown to induce effective immune responses and tumor regression [4]. Similarly, PD-1 was later found to be a negative immune regulator expressed on T-cells [5,6]. Its ligand, PD-L1, was shown to be expressed on normal tissues and tumor cells and to escape immune surveillance. The blockade of these molecules with monoclonal antibodies, such as ipilimumab (anti–CTLA-4), nivolumab (anti–PD-1), pembrolizumab (anti–PD-1), atezolizumab (anti-PD-L1), and avelumab (anti-PD-L1) has resulted in long-lasting responses in several solid tumors [7–12].

Oncolytic virus therapies use genetically modified viruses to infect tumor cells and enable the immune system to detect them. Oncolytic viruses have direct cytotoxic activity and result in epitope spreading and a proinflammatory microenvironment upon cell lysis, which further primes antitumor immunity [13].

Cancer vaccines activate T-cell–mediated responses against tumor cells via tumor-specific antigens. Cancer vaccines may be composed of whole tumor cells, peptides, DNA, or RNA [14].

Dendritic cells, considered to be the immune system's most effective antigen-presenting cells, can be combined with tumor antigens or tumor cell lysates followed by ex vivo stimulation and maturation. The mature dendritic cells are then administered to effect cytotoxic T lymphocyte and natural killer (NK) cell activity against tumor cells [15].

Cytokines are biochemical messengers released by cells in response to infection, inflammation, tumorigenesis, and other cellular stresses, in order to elicit coordinated immune responses to the target tissue. Cytokine-based cancer immunotherapies include IL-2 and interferon-alpha, which promote antitumor immune responses through cytotoxic T-cell growth and activation and the induction of tumor apoptosis and senescence, respectively [16].

Adoptive cell therapies—which include lymphokine-activated killer (LAK) cells, chimeric antigen receptor (CAR) T cells, T-cell receptor (TCR) engineered T cells, and NK cells—are used to target and eliminate cancer cells [17–20]. In the 1980s, LAK cell therapy was one of the earliest adoptive cell therapies and included the ex vivo expansion of NK and T cells—typically derived from peripheral blood mononuclear cells—via activation with IL-2 [21,22]. LAK cell therapy's use has since been discontinued because of its toxicity and low inherent tumor cytotoxicity [21,23,24]. NK cells can be modified with CARs or TCRs for tumor cell targeting and elimination via the release of cytolytic granules and cytotoxic cytokines [17,25]. CAR T-cells use antibody fragments that are engineered to detect specific antigens expressed on the surface of cancer cells independent of major histocompatibility complexes (MHC) or antigen-presenting cells [19]. In contrast, TCR engineered T cells use modified TCRs to recognize cancer antigen fragments bound to MHC molecules on antigen-presenting cells [20].

## 2. Rare Central Nervous System Tumors

### 2.1. Meningioma

Meningiomas are the most common type of CNS tumor (>37%) and comprise a histologically and molecularly heterogeneous group of tumors [26,27]. Atypical (WHO grade 2) and anaplastic (WHO grade 3) meningiomas make up 5–7% and 3–5% of all meningiomas, respectively [28]. Although some atypical or anaplastic meningiomas can be surveilled, a small subset of these tumors recur without effective treatment options and no systemic treatments for recurrence have been approved by the FDA. The current landscape of immunotherapy studies has identified several potential immune checkpoint targets for recurrent meningioma, including PD-1, PD-L1, PD-L2, and B7-H3. PD-L1 was shown to be highly expressed in some meningiomas and was found to be associated with high tumor grades [29–32], suggesting a potential role for treating meningioma with ICIs. Avelumab combined with engineered NK cell lines demonstrated efficacy in tumor cell lysis in vitro [33]. Other targets of interest are M2 macrophages, which promote an immunosuppressive tumor microenvironment. In a murine meningioma model, the blockade of the colony-stimulating factor 1 (CSF1)-receptor

pathway, which is involved in M2 macrophage differentiation, restricted tumor growth [34]. Other checkpoint proteins expressed include PD-L2 and B7-H3 in meningiomas with alterations in the PI3K/AKT/mTOR pathway genes [35], CTLA-4 in CD3+ T cells in atypical meningiomas with PIK3CA or smoothened, frizzled class receptor (SMO) mutations [35], and the cancer/testis antigen NY-ESO-1 [36,37].

Few clinical trials have investigated checkpoint inhibition in atypical and anaplastic meningiomas. Although nivolumab was well-tolerated in a recent phase 2 trial in patients with recurrent atypical or anaplastic meningiomas, the drug did not increase 6-month progression-free survival (PFS) rates in comparison with historical benchmarks [38]. Another phase 2 trial of pembrolizumab in high-grade meningiomas met its primary endpoint in improving 6-month PFS rates with a median PFS of 7.6 months in comparison to historical controls [39]. Interim results of a phase 1/2 study of stereotactic radiosurgery plus nivolumab with or without ipilimumab for high-grade meningiomas were reported for 13 patients. Despite both regimens being well tolerated, five patients experienced disease progression and four died after a median follow-up of 11.1 months [40]. However, a report of a patient with a mismatch repair protein (MutS homolog 2 (MSH2))-deficient meningioma showed robust antitumor activity in response to nivolumab [41]. Ongoing clinical trials in meningioma include ICIs and TCR T-cell immunotherapies (Table 1).

**Table 1.** Ongoing clinical trials of immunotherapy for brain tumors.

| Cancer Type | Trial Number | Sponsor | Trial Name | Recruitment Status (as of 4/18/23) | Locations |
|---|---|---|---|---|---|
| Meningioma | NCT03604978 | National Cancer Institute (NCI) | Nivolumab and Multi-fraction Stereotactic Radiosurgery with or without Ipilimumab in Treating Patients with Recurrent Grade II-III Meningioma | Recruiting | US |
| | NCT04659811 | University of California, San Francisco | Stereotactic Radiosurgery and Immunotherapy (Pembrolizumab) for the Treatment of Recurrent Meningioma | Recruiting | US |
| | NCT02648997 | Dana-Farber Cancer Institute | An Open-Label Phase II Study of Nivolumab in Adult Participants with Recurrent High-Grade Meningioma | Recruiting | US |
| | NCT04728568 | Beijing Tiantan Hospital | Exploratory Study of PD-1 Neoadjuvant Treatment of Recurrent Meningioma | Recruiting | China |
| | JPRN-UMIN000036642 | Keio University School of Medicine | Accompanying Study in a Physician-Initiated Phase II Clinical Trial of Anti-PD-1 Antibody Therapy for Relapsed/Progressive Meningioma | Recruiting | Japan |
| | NCT03279692 | Massachusetts General Hospital | Phase II Trial of Pembrolizumab in Recurrent or Residual High-Grade Meningioma | Active, not recruiting | US |
| | NCT03267836 | Washington University School of Medicine | Neoadjuvant Avelumab and Hypofractionated Proton Radiation Therapy Followed by Surgery for Recurrent Radiation-Refractory Meningioma | Active, not recruiting | US |
| | NCT01967823 | National Cancer Institute (NCI) | T Cell Receptor Immunotherapy Targeting NY-ESO-1 for Patients with NY-ESO-1 Expressing Cancer | Completed, results pending | US |
| Pituitary Carcinoma | NCT04042753 | Memorial Sloan Kettering Cancer Center | Nivolumab and Ipilimumab in People with Aggressive Pituitary Tumors | Recruiting | US |
| Ependymoma | NCT01795313 | Ian F. Pollack, M.D., University of Pittsburgh | Immunotherapy for Recurrent Ependymomas in Children Using Tumor Antigen Peptides with Imiquimod | Recruiting | US |
| | NCT04903080 | Pediatric Brain Tumor Consortium | HER2-Specific Chimeric Antigen Receptor (CAR) T Cells for Children with Ependymoma | Active, not recruiting | US |
| | NCT04408092 | University of Colorado, Denver | Study of the Effect of GM-CSF on Macrophages in Ependymoma | Active, not recruiting | US |

**Table 1.** *Cont.*

| Cancer Type | Trial Number | Sponsor | Trial Name | Recruitment Status (as of 4/18/23) | Locations |
|---|---|---|---|---|---|
| Medulloblastoma | NCT02962167 | Sabine Mueller, M.D., PhD, University of California, San Francisco | Modified Measles Virus (MV-NIS) for Children and Young Adults with Recurrent Medulloblastoma or Recurrent ATRT | Recruiting | US |
| | NCT03299309 | Eric Thompson, M.D., Duke University | PEP-CMV in Recurrent Medulloblastoma/Malignant Glioma (PRiME) | Active, not recruiting | US |
| | NCT01326104 | University of Florida | Vaccine Immunotherapy for Recurrent Medulloblastoma and Primitive Neuroectodermal Tumor (Re-MATCH) | Active, not recruiting | US |
| | NCT04167618 | Y-mAbs Therapeutics Inc. (New York, NY, USA) | 177Lu-DTPA-Omburtamab Radioimmunotherapy for Recurrent or Refractory Medulloblastoma | Terminated (business priorities) | Denmark, Netherlands, Spain, United Kingdom, US |
| | NCT02332889 | University of Louisville | Phase I/II: Decitabine/Vaccine Therapy in Relapsed/Refractory Pediatric High-Grade Gliomas/Medulloblastomas/CNS PNETs | Terminated (transition to a different immunotherapy strategy in the future at our institution) | US |
| Atypical Teratoid/Rhabdoid Tumors | NCT04416568 | Dana-Farber Cancer Institute | Study of Nivolumab and Ipilimumab in Children and Young Adults with INI1-Negative Cancers | Recruiting | US |
| | NCT02962167 | Sabine Mueller, M.D., PhD, University of California, San Francisco | Modified Measles Virus (MV-NIS) for Children and Young Adults with Recurrent Medulloblastoma or Recurrent ATRT | Recruiting | US |
| | NCT05286801 | National Cancer Institute (NCI) | Tiragolumab and Atezolizumab for the Treatment of Relapsed or Refractory SMARCB1 or SMARCA4 Deficient Tumors | Recruiting | US |

**Table 1.** *Cont.*

| Cancer Type | Trial Number | Sponsor | Trial Name | Recruitment Status (as of 4/18/23) | Locations |
|---|---|---|---|---|---|
| Glioma | NCT05106296 | Theodore S. Johnson, Augusta University | Chemo-Immunotherapy Using Ibrutinib Plus Indoximod for Patients with Pediatric Brain Cancer | Recruiting | US |
| | NCT04978727 | Pediatric Brain Tumor Consortium | A Pilot Study of SurVaxM in Children Progressive or Relapsed Medulloblastoma, High Grade Glioma, Ependymoma, and Newly Diagnosed Diffuse Intrinsic Pontine Glioma | Recruiting | US |
| | NCT04661384 | City of Hope Medical Center | Brain Tumor-Specific Immune Cells (IL13Ralpha2-CAR T Cells) for the Treatment of Leptomeningeal Glioblastoma, Ependymoma, or Medulloblastoma | Recruiting | US |
| | NCT04185038 | Seattle Children's Hospital | Study of B7-H3-Specific CAR T Cell Locoregional Immunotherapy for Diffuse Intrinsic Pontine Glioma/Diffuse Midline Glioma and Recurrent or Refractory Pediatric Central Nervous System Tumors | Recruiting | US |
| | NCT04049669 | Theodore S. Johnson, Augusta University | Pediatric Trial of Indoximod with Chemotherapy and Radiation for Relapsed Brain Tumors or Newly Diagnosed DIPG | Recruiting | US |
| | NCT03911388 | University of Alabama at Birmingham | HSV G207 in Children with Recurrent or Refractory Cerebellar Brain Tumors | Recruiting | US |
| | NCT03500991 | Seattle Children's Hospital | HER2-Specific CAR T Cell Locoregional Immunotherapy for HER2-Positive Recurrent/Refractory Pediatric CNS Tumors | Recruiting | US |
| | NCT03173950 | National Cancer Institute (NCI) | Immune Checkpoint Inhibitor Nivolumab in People with Recurrent Select Rare CNS Cancers | Recruiting | US |
| | NCT03152318 | Dana-Farber Cancer Institute | A Study of the Treatment of Recurrent Malignant Glioma with rQNestin34.5v.2 (rQNestin) | Recruiting | US |
| | NCT02359565 | National Cancer Institute (NCI) | Pembrolizumab in Treating Younger Patients with Recurrent, Progressive, or Refractory High-Grade Gliomas, Diffuse Intrinsic Pontine Gliomas, Hypermutated Brain Tumors, Ependymoma, or Medulloblastoma | Recruiting | Canada, US |

| Cancer Type | Trial Number | Sponsor | Trial Name | Recruitment Status (as of 4/18/23) | Locations |
|---|---|---|---|---|---|
| Glioma | JPRN-UMIN000029005 | Keio University School of Medicine | VEGFR1/2 Peptide Vaccine in Patients with Recurrent, Progressive, and Refractory Brain Tumors | Recruiting | Japan |
| | EUCTR2020-004838-37-ES | Fundación de Investigación Biomédica Hospital Niño Jesús | Phase IB Clinical Trial to Assess the Safety, Tolerability, and Preliminary Efficacy of AloCELYVIR in Children, Adolescents, and Young Adults with Diffuse Intrinsic Pointine Glioma (DIPG) or Medulloblastoma | Recruiting | Spain |
| | NCT03615404 | Gary Archer Ph.D., Duke University | Cytomegalovirus (CMV) RNA-Pulsed Dendritic Cells for Pediatric Patients and Young Adults with WHO Grade IV Glioma, Recurrent Malignant Glioma, or Recurrent Medulloblastoma (ATTAC-P) | Completed | US |
| | NCT02834013 | National Cancer Institute (NCI) | Nivolumab and Ipilimumab in Treating Patients with Rare Tumors | Active, not recruiting | US |
| | NCT03389802 | Pediatric Brain Tumor Consortium | Phase I Study of APX005M in Pediatric CNS Tumors | Active, not recruiting | US |
| | NCT03043391 | Istari Oncology, Inc. (Durham, NC, US) | Phase 1b Study PVSRIPO for Recurrent Malignant Glioma in Children | Active, not recruiting | US |
| | NCT02457845 | University of Alabama at Birmingham | HSV G207 Alone or with a Single Radiation Dose in Children with Progressive or Recurrent Supratentorial Brain Tumors | Active, not recruiting | US |
| | NCT02444546 | Mayo Clinic | Wild-Type Reovirus in Combination with Sargramostim in Treating Younger Patients with High-Grade Relapsed or Refractory Brain Tumors | Active, not recruiting | US |
| | NCT02100891 | Monica Thakar, Medical College of Wisconsin | Phase 2 STIR Trial: Haploidentical Transplant and Donor Natural Killer Cells for Solid Tumors (STIR) | Active, not recruiting | US |
| | NCT03638167 | Seattle Children's Hospital | EGFR806-specific CAR T Cell Locoregional Immunotherapy for EGFR-Positive Recurrent or Refractory Pediatric CNS Tumors | Active, not recruiting | US |
| | NCT00634231 | Candel Therapeutics, Inc. (Needham, MA, US) | A Phase I Study of AdV-tk + Prodrug Therapy in Combination with Radiation Therapy for Pediatric Brain Tumors | Completed, no results posted | US |

**Table 1.** *Cont.*

| Cancer Type | Trial Number | Sponsor | Trial Name | Recruitment Status (as of 4/18/23) | Locations |
|---|---|---|---|---|---|
| Glioma | NCT01082926 | City of Hope Medical Center | Phase I Study of Cellular Immunotherapy for Recurrent/Refractory Malignant Glioma Using Intratumoral Infusions of GRm13Z40-2, an Allogeneic CD8+ Cytolitic T-Cell Line Genetically Modified to Express the IL 13-Zetakine and HyTK and to be Resistant to Glucocorticoids, in Combination with Interleukin-2 | Completed, no results posted | US |
| | NCT02502708 | NewLink Genetics Corporation (Ames, IA, US) | Study of the IDO Pathway Inhibitor, Indoximod, and Temozolomide for Pediatric Patients with Progressive Primary Malignant Brain Tumors | Completed, no results posted | US |
| | NCT00730613 | City of Hope Medical Center | Cellular Adoptive Immunotherapy Using Genetically Modified T-Lymphocytes in Treating Patients with Recurrent or Refractory High-Grade Malignant Glioma | Completed, no results posted | US |
| | NCT01171469 | Masonic Cancer Center, University of Minnesota | Vaccination with Dendritic Cells Loaded with Brain Tumor Stem Cells for Progressive Malignant Brain Tumor | Completed, no results posted | US |
| | NCT00014573 | Barbara Ann Karmanos Cancer Institute | Chemotherapy and Vaccine Therapy Followed by Bone Marrow or Peripheral Stem Cell Transplantation and Interleukin-2 in Treating Patients with Recurrent or Refractory Brain Cancer | Completed, no results posted | US |
| | NCT04730349 | Bristol-Myers Squibb (New York, NY, US) | A Study of Bempegaldesleukin (BEMPEG: NKTR-214) in Combination with Nivolumab in Children, Adolescents, and Young Adults with Recurrent or Treatment-resistant Cancer (PIVOT IO 020) | Terminated (business objectives have changed) | Australia, France, Germany, Italy, Spain, US |

Abbreviations: NCI, National Cancer Institute; US, United States.

### 2.2. Pituitary Carcinoma and Aggressive Pituitary Adenoma

Pituitary tumors originate from the endocrine cells of the anterior pituitary and account for about 15% of all intracranial neoplasms. Although most pituitary tumors are considered benign, about 0.1% of all pituitary tumors can display aggressive behavior and metastasize and are classified as pituitary carcinoma (PC) [42]. PCs require multiple lines of treatment including surgery, radiation, and chemotherapy. Such treatment is often inadequate in controlling the disease, and the mean survival time is usually less than 4 years [43]. Much interest has been shown in ICIs for the management of PCs and aggressive pituitary adenomas. The rationale for ICI use in these tumors is based on several findings. First, PCs contain tumor-infiltrating lymphocytes [44–47] and express PD-L1, which is suggested to be a predictor of the response to ICIs [44,46–48]. Second, autoimmune hypophysitis is a known immune-related adverse event of ICIs caused by overactive lymphocytes. Although the exact pathogenesis is unknown, CTLA-4 and PD-1 expression in pituitary cells are thought to play a role [49]. Third, a PD-L1 blockade with ICIs reduced ACTH levels and tumor growth and increased survival in murine models [44]. Finally, there are an increasing number of case reports and case series demonstrating ICI effectiveness in PC and aggressive pituitary adenoma [50].

Several recent case reports have described immunotherapy responses in PCs and aggressive pituitary adenomas (Table 2). Lin et al. reported a response to the combination of ipilimumab and nivolumab in a patient with ACTH-secreting PC [51]. The patient subsequently had recurrences which required surgery, external beam radiation, and Lu-DOTATATE but had a continued response that was attributed to ipilimumab and nivolumab for 3.5 years [52]. In a report of two patients treated with ipilimumab plus nivolumab, partial biochemical and radiographic responses were observed in the patient who had ACTH-secreting PC, but not in the one who had prolactin-secreting PC [53]. In a phase 2 study of pembrolizumab in rare malignancies (NCT02721732), partial responses occurred in two out of four patients who had ACTH-secreting PC [54], but not in the patient who had a non-secreting corticotroph PC, nor the one who had a prolactin-secreting PC. In another report, a patient with a non-secreting lactotroph PC was treated with ipilimumab and nivolumab and demonstrated a clinical and radiographic response that was sustained for 8 months [55]. One patient with a prolactin-secreting PC experienced a significant biochemical and radiological improvement that persisted for 24 months with ipilimumab and nivolumab [56]. Another patient with an ACTH-secreting pituitary adenoma showed biochemical improvement and radiological stabilization with ipilimumab and nivolumab treatment, which persisted for 1 year after ICI initiation [57]. The cases reported by Lin, Lamb, Majd, and their colleagues revealed hypermutator phenotypes, which included mutations in mismatch repair (MMR) genes that were attributed to prior temozolomide treatment [52,54,55]. Among the patients described in these reports, PD-L1 expression was elevated in the tumor of only one of six responders and was unknown in one case. Immunotherapy was tolerated well except in two patients: one who experienced asthenia, anorexia, and progressive weight loss and one with nausea, vomiting, and grade 3 diarrhea [53]. Although these cases reported positive responses to immunotherapy, one patient with an ACTH-secreting PC with DNA MMR deficiency had rapid progression after treatment with pembrolizumab [58]. Currently, two trials are investigating ICIs in PCs and aggressive pituitary adenomas (Table 1).

**Table 2.** Pituitary carcinoma case report summaries.

| Initial Tumor Diagnosis | Time (Months) from PA to PC Diagnosis | PDL-1 Expression | Number of Prior Surgeries | Number of Prior Radiation Treatment Courses | Number of Prior Chemotherapy Regimens | Immunotherapy Regimen | Radiographic Response | Biochemical Response | PFS (Months) after ICI Initiation | Clinical Outcome | Reference |
|---|---|---|---|---|---|---|---|---|---|---|---|
| Corticotroph pituitary adenoma | 68 | <1% | 4 | 2 | 2 | Ipilimumab/nivolumab (5 cycles), maintenance nivolumab | Yes | Yes | 8 | Alive 30 months at end of study period after PC diagnosis. | Lin et al. 2018 [51] |
| Corticotroph pituitary adenoma | 68 | <1% | 4 | 2 | 3 | Ipilimumab/nivolumab (5 cycles), maintenance nivolumab (ongoing) | Yes | Yes | 8 | Alive 30 months at end of study period after PC diagnosis. | Lin et al. 2021 [52] |
| Corticotroph (silent) pituitary adenoma | 216 | <1% | 2 | 1 | 1 | Pembrolizumab (4 cycles) | Yes | No | 4 | Transitioned to fotemustine and alive 6 months at end of study. | Caccese et al. 2020 [58] |
| Corticotroph pituitary adenoma | 205 | <1% | 3 | 2 | 2 | Ipilimumab/nivolumab (5 cycles), maintenance nivolumab (12 cycles) | Yes | Yes | 5 | Worsening progression 12 months after ICI initiation followed by death at 14 months from unknown cause. | Duhamel et al. 2020 [53] |
| Lactotroph pituitary adenoma | 88 | Unknown | 3 | 1 | 1 | Ipilimumab/nivolumab (2 cycles) | No | No | 0 | Rapid progression after 2 cycles and transitioned to bevacizumab with prolactin level stability. | Duhamel et al. 2020 [53] |
| Lactotroph (silent) pituitary adenoma | 45 | <1% | 4 | 2 | 1 | Ipilimumab/nivolumab (2 cycles), maintenance nivolumab (17 cycles), ipilimumab/nivolumab re-challenge (4 cycles) | Yes | N/A | 8 | After progression, re-challenge with ipilimumab/nivolumab (4 cycles) had no response. Experienced auto-immune nephritis with both courses of ipilimumab/nivolumab and treated with corticosteroids. | Lamb et al. 2020 [55] |
| Corticotroph pituitary adenoma | 54 | <1% | 3 | 8 | 6 | Pembrolizumab (29 cycles) | Yes | Yes | 69 | No progression at end of study. | Majd et al. 2020 [54] |
| Corticotroph pituitary adenoma | 45 | <1% | 2 | 1 | 2 | Pembrolizumab (34 cycles) | Yes | Yes | 32 | No progression at end of study. | Majd et al. 2020 [54] |
| Corticotroph (silent) pituitary adenoma | 131 | <1% | 3 | 4 | 4 | Pembrolizumab (6 cycles) | Yes (stable) | N/A | 4 | Alive 138 months after PC diagnosis. | Majd et al. 2020 [54] |
| Lactotroph pituitary adenoma | 81 | <1% | 1 | 2 | 4 | Pembrolizumab (6 cycles) | No | No | 0 | Deceased 46 months after PC diagnosis. | Majd et al. 2020 [54] |

**Table 2.** *Cont.*

| Initial Tumor Diagnosis | Time (Months) from PA to PC Diagnosis | PDL-1 Expression | Number of Prior Surgeries | Number of Prior Radiation Treatment Courses | Number of Prior Chemotherapy Regimens | Immunotherapy Regimen | Radiographic Response | Biochemical Response | PFS (Months) after ICI Initiation | Clinical Outcome | Reference |
|---|---|---|---|---|---|---|---|---|---|---|---|
| Lactotroph pituitary adenoma | 81 | 95% | 2 | 3 | 1 | Ipilimumab/nivolumab (4 cycles), maintenance nivolumab | Yes | Yes | 24 | No progression at end of study. | Goichot et al. 2021 [56] |
| Corticotroph pituitary adenoma | 76 | Unknown | 2 | 2 | 1 | Ipilimumab/nivolumab (4 cycles) followed by maintenance nivolumab | Yes (stable) | Yes | 12 | No progression at end of study. | Sol et al. 2021 [57] |

Abbreviations: ICI, immune checkpoint inhibitor; N/A, not applicable; PA, pituitary adenoma; PC, pituitary carcinoma; PFS, progression-free survival.

### 2.3. Ependymoma

Ependymomas are gliomas that account for around 4% of all primary CNS tumors and occur more often in children than in adults. Ependymomas usually have low cell density and a low mitotic index [59]. Patients with these tumors have varying clinical outcomes, which mainly depend on molecular subgroups [60]. Mounting evidence shows that a subset of recurrent ependymomas have an immunosuppressive phenotype [61,62] associated with T-cell exhaustion [63]. PD-1 and PD-L1 expression have been observed in supratentorial ependymomas [64], posterior fossa ependymomas, and myeloid cells [63,65,66]; the highest expression levels were seen in supratentorial RELA fusion–positive ependymomas [65,66]. Furthermore, PD-1 was found to be expressed on infiltrating CD4+ and CD8+ T cells [63]. An increase in tumor mutational burden and neoantigen load have also been observed in ependymoma after multiple treatments [67]. Additionally in a murine xenograft model of metastatic ependymoma, the cell surface markers EPHA2, Il-13Ra2, and HER2 were identified and targeted with CAR T cells, demonstrating promising therapeutic responses [68].

The aforementioned studies indicate that ICIs are valuable therapeutic modalities and few case reports have described ICI responses in recurrent ependymomas. Anti–PD-1 therapy (tislelizumab (BGB-A317)) demonstrated stable disease after more than 18 months in a patient with metastatic myxopapillary ependymoma [69], which was a longer PFS than that previously reported with systemic therapies for ependymoma. PD-L1 expression was seen on 0% of tumor cells and 5% of tumor-infiltrating immune cells, indicating that the response to ICI treatment was independent of PD-L1 expression [69]. One pediatric patient with recurrent RELA-fusion, PD-L1+ (20%) ependymoma was treated with nivolumab and sirolimus and had stable disease for 1 year after the initiation of immunotherapy [70]. Another patient with spinal ependymoma treated with ipilimumab and nivolumab for 18.6 months had stable disease [71].

Multiple immunotherapy clinical trials for patients with recurrent ependymoma are recently completed or ongoing. A phase 1 study with oncolytic viral therapy with aglatimagene besadenovec included one patient with recurrent ependymoma, who had no serious adverse events, no dose-limiting toxicities, and no progression at the last reported follow-up of 47.7 months [72]. A phase 1b/2 trial (Checkmate 908) that included 22 pediatric patients with relapsed/resistant ependymoma was recently completed, and the preliminary data demonstrated no clinical benefit with a median PFS of 1.4 months (range, 1.4–2.6) with nivolumab (n = 12) and 4.6 months (range, 1.4–5.4) with nivolumab and ipilimumab (n = 10) [73]. Another recent phase 1 study of intraventricular autologous NK cells for pediatric recurrent ependymoma had no dose-limiting toxicities, and one of nine patients showed stable disease but was taken off the study early per parent preference [74]. A previous phase 1 study, which used dendritic cells loaded with a glioblastoma cell line (GBM6-AD), included one patient with posterior fossa ependymoma. The treatment was well tolerated; however, the patient developed spinal metastasis 20 weeks after vaccination [75]. Other ongoing trials of immunotherapy for recurrent ependymoma include CAR T cells, ICIs, cytokines, and oncolytic viruses (Table 1).

### 2.4. Medulloblastoma

Medulloblastoma (MB), an embryonal tumor of the posterior fossa, is the most common malignant brain cancer in children. MB is uncommon in adults; however, there are about 140 new cases in adults in the United States per year [76–78]. Adult MBs are heterogeneous in nature, with several molecular subgroups, and are known to have an immunosuppressive tumor environment [1,76,79]. The most common molecular subgroup of adult MB is Sonic Hedgehog (SHH), followed by group 4 and Wingless. The current conventional management is often multimodal and includes the maximum safe resection followed by craniospinal radiotherapy with or without concurrent and/or adjuvant chemotherapy [80]. The treatment options for recurrent MB are limited. Several current immunotherapy approaches for the management of MB include vaccines, oncolytic viruses, checkpoint inhibitors, NK cells, and CAR T cells [81].

Cancer vaccine use in MB is under active investigation. A recent phase 1 study that used a peptide vaccine directed to CMV pp65 for the treatment of recurrent glioma and medulloblastoma demonstrated good tolerance and elicited an immune response in heavily pretreated recurrent patients and will be analyzed further in an upcoming phase 2 study (NCT05096481) [82]. The preliminary results from a phase 1 study of cytomegalovirus RNA–pulsed dendritic cells did not show significant adverse events or dose-limiting toxicity in all 11 patients (NCT03615404). RNA-loaded autologous dendritic cells from patients with MB were successfully generated [83] and are currently being studied in the phase 2 Re-MATCH trial (NCT01326104). Prior cancer vaccines, including autologous dendritic and tumor cells, have largely been unsuccessful in adult patients with MB (NCT02332889, NCT01171469, and NCT00014573).

Oncolytic viruses for MB remain a work in progress. Thompson and colleagues used a recombinant poliovirus rhinovirus (PVSRIPO) in vitro to target the poliovirus receptor CD155 in MB cell lines (D283, D341). In this study (NCT03043391), PVSRIPO was capable of propagating and of infecting, prohibiting cell proliferation by, and killing group 3 MB [84]. Other oncolytic viruses for MB under investigation include the human synthetic cytomegalovirus matrix protein pp65 vaccine (NCT03299309), modified measles (NCT02962167), reovirus (NCT02444546), and herpes simplex virus-1 (NCT03911388).

PD-L1 expression has been variably reported in medulloblastoma. In some studies, PD-L1 expression in MB has been shown to be low or absent [64,85,86]. In contrast, PD-L1 expression was found to be highest in SHH MB but varied among other MB subgroups [85]. Despite the low expression of PD-L1 in human MB, a PD-1 blockade in murine MB models showed greater antitumor efficacy in group 3 MB than in SHH MB [87]. A recent phase 1b/2 study (Checkmate 908) that included 30 pediatric patients with relapsed/resistant MB showed a median PFS of 1.4 months (range, 1.2–1.4) with nivolumab (n = 15) and 2.8 months (range, 1.5–4.5) with nivolumab and ipilimumab (n = 15) [73]. As mentioned earlier, this study did not show any significant benefit from nivolumab and ipilimumab treatment for high-grade pediatric malignancies. Several ICIs are under investigation in clinical trials for MB treatment, including nivolumab (NCT03173950), pembrolizumab (NCT02359565), and durvalumab (NCT02793466).

Adoptive cell immunotherapy with NK cells has been shown to target MB cells through the activation of the NK group 2 member D (NKG2D) activator receptor via the ligands on MB cells [88]. Blocking the NKG2D receptor on NK cells and ligands (MICA/ULBP-2) on an MB cell line (HTB-186) increased the resistance to NK cell-mediated lysis in vitro [89]. Additionally, MB cell lines (Daoy, Med8A) and human MB tumors express CD1d, an antigen-presenting molecule for NK cells that may represent another target for MB immunotherapy [90]. LAK cells were also found to target MB cells in vitro [91,92] and in a case series of eight patients who had MB with cerebrospinal fluid dissemination, LAK cells were injected intrathecally for 3 months and resulted in neurological improvements and complete responses that lasted as long as 20 months in three patients [93]. One MB patient treated with intrathecal LAK cells experienced disease progression with 40-week survival after immunotherapy [94]. In another study, intrathecal LAK cell therapy in two patients with disseminated MB was not successful [95]. A phase 1 study by Khatua and colleagues demonstrated the safety and feasibility of the ventricular infusion of autologous NK cells in patients with recurrent posterior fossa tumors, including five pediatric patients with MB [74]. One patient with MB had stable disease after five infusions but eventually experienced disease progression, as did the other four MB patients. Donor NK cell administration after allogenic hematopoietic cell transplantation and reduced-intensity radiotherapy and chemotherapy are being studied in a phase 2 clinical trial (NCT02100891).

Preclinical studies have identified MB target antigens, including HER2 and B7-H3 [96,97]. HER2-CAR T cells cleared MB via intraventricular and intravenous delivery in mouse xenograft models without significant toxicity [97]. B7-H3 CAR T cells caused in vivo tumor regression in murine xenograft models [96]. Currently, EGFR-specific CAR T cell therapy for MB is under investigation in a phase 1 study (NCT03638167).

### 2.5. Atypical Teratoid/Rhabdoid Tumors

Atypical teratoid/rhabdoid tumors (ATRT) originate from the biallelic inactivation of SMARCB1, a component of the switch/sucrose nonfermentable complex, which is a major regulator of chromatin remodeling. ATRTs typically occur in children aged 3 years or younger and make up 1–2% of all pediatric CNS tumors and are less common in adults [2,98]. These tumors are known to have a low mutational burden [99,100], and PD-L1 expression has varied among the reported studies [99,101]. However, a growing body of evidence demonstrates that these tumors are highly immunogenic [100,102,103], which suggests that immunotherapy may be an option for the management of these tumors. One proposed immunogenic mechanism is splicing disruption [104,105], mediated by the dysfunction of the switch/sucrose nonfermentable complex, to generate neoepitopes. Additionally, interference with the epigenetic silencing of endogenous retroviruses may also trigger immunogenicity from the defects of SMARCB1, which regulates the sense and antisense expression of retrovirus loci in DNA [100]. ATRTs were also found to express the B7-H3 antigen, which is expressed primarily in prenatal but not postnatal brain cells and is a target of current clinical trials [106]. Intrathecal CAR T cells targeting B7-H3 led to tumor regression in a patient-derived xenograft murine model [106].

Immunotherapy in clinical trials for patients with ATRT has not been successful to date. A phase 1/2 study of atezolizumab in children and young adults that included three patients with ATRTs was conducted. Of these patients, two had progressive disease and one had missing data after a median follow-up time of 6.8 months [107]. None of these patients had PD-L1 expression. Another phase 1/2 trial of pembrolizumab in pediatric patients showed disease progression in three out of four patients with ATRTs after a median follow-up of 8.6 months. Of all the screened ATRT patients in the pembrolizumab study, PD-L1 expression was detected in 10 out of 16 (63%) [101]. Current clinical trials for ATRT treatment include CAR T cells, ICIs, and oncolytic virus therapies (Table 1).

### 2.6. Solitary Fibrous Tumors

Solitary fibrous tumors (SFTs) are rare mesenchymal tumors with locally invasive properties. SFTs most commonly involve the pleura and lung followed by the meninges [108]. The incidence is less than one case per million people per year and they affect adult patients in the sixth decade of life [109]. Meningeal SFTs were previously called "hemangiopericytoma" but the term was removed from the recent 2021 WHO Classification of Tumors of the Central Nervous System nomenclature [1]. Tumorigenesis is driven through the NAB2-STAT6 fusion protein, which causes the downstream activation of the MAPK/ERK pathway through a positive feedback loop and is detected in nearly all SFTs [110–112]. Meningeal SFTs have a poor prognosis with frequent recurrence and brain parenchymal and calvarial invasion despite multimodal management including surgery, radiotherapy, chemotherapy, and targeted therapies.

Few studies have investigated the use of immunotherapy in meningeal SFTs. SFTs have been shown to frequently express PD-L1, which positively correlated to the occurrence of extracranial metastases [113]. In contrast, Dancsok and colleagues found low expression of PD-L1 in 16 SFT cases [114]. Meningeal SFTs demonstrated high proportions of tumor-infiltrating lymphocytes (91.7%) but had low PD-L1 expression (8.3%) [115]. A phase 2 study investigating pembrolizumab in sarcomas showed PD-L1 expression in 12% and 40% of tumor cells and infiltrating immune cells, respectively. Of all 50 patients in this study, only one patient had an SFT and had a partial response to pembrolizumab and was progression free at 6 months [116]. A potential new immunotherapy target in SFTs includes preferentially expressed antigen in melanoma (PRAME), which is seen in 58% of SFTs and is thought to play a role in immune evasion, a decreased proportion of antigen presenting cells, and the expression of the anti-phagocytic tumor cell marker CD47 [117]. To our knowledge, as of the time of writing, there are no immunotherapy trials open for meningeal SFTs.

## 3. Conclusions

Patients with rare CNS tumors represent a population that often has limited treatment options, especially in the recurrent setting. Immunotherapy serves as a possible management option and is undergoing rapid development. To date, immunotherapy studies have shown some promise with PD-1 and PD-L1 inhibitors in the treatment of meningiomas and pituitary carcinomas. Follow-up investigations are ongoing for ependymoma, medulloblastoma, ATRTs and meningeal SFTs to determine the role of immunotherapy. Ongoing clinical trials will help further our understanding and the development of immunotherapy for optimal patient care.

**Author Contributions:** Conceptualization, A.R. and N.K.M.; investigation, A.R., N.K.M. and C.K.-M.; writing—original draft preparation, A.R. and N.K.M.; writing—review and editing, A.R., N.K.M. and C.K.-M.; supervision, N.K.M. All authors have read and agreed to the published version of the manuscript.

**Funding:** This research received no external funding.

**Data Availability Statement:** Data is contained within the article.

**Acknowledgments:** Editorial support was provided by Bryan Tutt, Scientific Editor, Research Medical Library at MD Anderson Cancer Center.

**Conflicts of Interest:** The authors declare no conflict of interest.

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
