# Peer review of "The Role of Immunotherapy in the Treatment of Rare Central Nervous System Tumors"

_curroncol, doi:10.3390/curroncol30060401_

Round 1
Reviewer 1 Report
1. The title indicates adequately the study design.
2. The abstract provides an informative and balanced summary of the review.
3. In the manuscript there is an adequate and clear presentation and summary of the scientific literature.
4. The discussions are presented in a fluid, transparent and unbiased manner.
5. The table is exhaustive and complete.
Author Response
Reviewer 1 Comments:
“1. The title indicates adequately the study design.
- The abstract provides an informative and balanced summary of the review.
- In the manuscript there is an adequate and clear presentation and summary of the scientific literature.
- The discussions are presented in a fluid, transparent and unbiased manner.
- The table is exhaustive and complete.”
Response: Thank you so much and we appreciate your kind comments.
Reviewer 2 Report
Rodriguez at al. provide a thorough overview on ongoing trials and current evidence of immunotherapy in the treatment of rare tumors of the CNS.
Overall, despite of the focus of the paper, the manuscript seems well organized and informative.
I only have two comments: I was a bit confused regarding the provided table: many studies are listed under the cancer type "ependymoma" even though it does not only includes ependymoma. Would it be more correct to classify the studies under the cancer type "glioma"? (and is there a formatting issue in the table maybe? Or are some studies performed in different hospitals? I find it difficult to read- maybe more spaces between the listed studies?)
Secondly: only studies performed in the US have been included. If this is the aim of the authors, it should be stated in the methods and the title. if not, international studies should be included as well.
Author Response
Reviewer 2 Comments:
“Rodriguez at al. provide a thorough overview on ongoing trials and current evidence of immunotherapy in the treatment of rare tumors of the CNS.
Overall, despite of the focus of the paper, the manuscript seems well organized and informative.”
Response: We appreciate your nice comments and providing insight to improve the table.
“ /’I only have two comments: I was a bit confused regarding the provided table: many studies are listed under the cancer type "ependymoma" even though it does not only includes ependymoma. Would it be more correct to classify the studies under the cancer type "glioma"? (and is there a formatting issue in the table maybe? Or are some studies performed in different hospitals? I find it difficult to read- maybe more spaces between the listed studies?)/’”
Response: We have edited the table to clarify the trials among the tumor types. (Table 1)
“Secondly: only studies performed in the US have been included. If this is the aim of the authors, it should be stated in the methods and the title. if not, international studies should be included as well.”
Response: International studies have been included in the modified Table 1 (Table 1).
Reviewer 3 Report
Rodriguez et al summarize the current preclinical and clinical data of immunotherapies against rare CNS tumors. In general they give a great overview of the ongoing trials and the focus on mainly checkpoint inhibitors. A few minor corrections I would like to have addressed by the authors before considering for publication:
- line 93: "several potential immune checkpoint targets" please specify which targets
- line 98: another target are? M2 macrophages.
- line 101/102: cross 1 of the "in meningiomas" out
- line 103: SMO?
- line 115: MSH2, please specify abbreviation
- line 117: T cell, otherwise spelled with a hyphon "-"
- line 187/188: has ICI been tested in recurrent ependymomas, yet? or is that to be expected? you continue, it has been tested in lin 192 - please rephase, that it can go into 1 paragraph, otherwise it is jumping back and forth and it is difficult for the reader to follow. please restructure here.
-line 281: HER2-BBZ CAR I don't know whether everyone would understand the BBZ meaning, and whether this information is actually important here. either describe it a little more or write it out or leave this detail out.
- The conlusion can also give a better summary of which therapies worked nicely (a lot PD-L1 treatment) and which once should be followed up.
Author Response
Reviewer 3 Comments:
“Rodriguez et al summarize the current preclinical and clinical data of immunotherapies against rare CNS tumors. In general they give a great overview of the ongoing trials and the focus on mainly checkpoint inhibitors. A few minor corrections I would like to have addressed by the authors before considering for publication:”
Response: Thank you so much for your thoughtful comments to improve the manuscript.
“- line 93: "several potential immune checkpoint targets" please specify which targets”
Response: Included checkpoint targets (Line 94)
“- line 98: another target are? M2 macrophages.”
Response: Fixed (Line 98)
“- line 101/102: cross 1 of the "in meningiomas" out”
Response: Fixed (Line 102)
“- line 103: SMO?”
Response: Clarified gene name. (Line 104)
“- line 115: MSH2, please specify abbreviation”
Response: Abbreviation specified (Line 116-117).
“- line 117: T cell, otherwise spelled with a hyphon "-"”
Response: Added (Line 118).
“- line 187/188: has ICI been tested in recurrent ependymomas, yet? or is that to be expected? you continue, it has been tested in lin 192 - please rephase, that it can go into 1 paragraph, otherwise it is jumping back and forth and it is difficult for the reader to follow. please restructure here.”
Response: Restructured sentences (Lines 194-202)
“-line 281: HER2-BBZ CAR I don't know whether everyone would understand the BBZ meaning, and whether this information is actually important here. either describe it a little more or write it out or leave this detail out.”
Response: Fixed (Line 291).
“- The conlusion can also give a better summary of which therapies worked nicely (a lot PD-L1 treatment) and which once should be followed up.”
Response: Added more detail in conclusion (Lines 350-354).